# *Posidonia oceanica* (L.) Delile at Its Westernmost Biogeographical Limit (Northwestern Alboran Sea): Meadow Features and Plant Phenology

**Ángel Mateo-Ramírez** [1,*] , **Pablo Marina** [1] , **Alejandro Martín-Arjona** [1] , **Elena Bañares-España** [2] ,
**José E. García Raso** [3] , **José L. Rueda** [1] and **Javier Urra** [1]

1    Centro Oceanográfico de Málaga, Instituto Español de Oceanografía-Consejo Superior de Investigaciones Científicas, Puerto Pesquero s/n, 29640 Málaga, Spain
2    Departamento de Botánica y Fisiología Vegetal, Campus de Teatinos s/n, Universidad de Málaga, 29071 Málaga, Spain
3    Departamento de Biología Animal, Campus de Teatinos s/n, Universidad de Málaga, 29071 Málaga, Spain
*    Correspondence: angel.mateo@ieo.csic.es

**Abstract:** Meadows of the seagrass *Posidonia oceanica* inhabit most infralittoral bottoms of the Mediterranean Sea and are considered one of the main climax stages of the infralittoral environment. This seagrass has its western distributional limit along the coast of the Alboran Sea. Taking into account the decline of *P. oceanica* meadows and the global scenario of ocean warming, it becomes essential to know the structure, temporal dynamics, sexual reproduction and conservation status of this seagrass, across its geographical distribution, including the distribution boundaries where the meadows withstand limiting environmental conditions. In the present work, we studied the structure, phenology and flowering events of four *P. oceanica* meadows located in the northwestern Alboran Sea (close to the Strait of Gibraltar). Results indicate a decreasing trend of patch size, bathymetric range and number of leaves per shoot towards the Strait (and the Atlantic Ocean), as well as an increasing trend of shoot density and leaf height. Phenological parameters of the northwestern Alboran Sea *P. oceanica* meadows presented temporal dynamics similar to meadows from other locations within the biogeographical distribution of this seagrass, with similar or even less annual variability in the former. Although most of the studied *P. oceanica* meadows seem to present a good health status (BiPo index ~0.6) with high shoot densities and some flowering events, some of them showed evidence of regression.

**Keywords:** seagrass; Mediterranean Sea; Atlantic influence; life cycle; infralittoral; habitat regression

## 1. Introduction

*Posidonia oceanica* (L) Delile is a Mediterranean endemic seagrass and, in optimal conditions, is considered the main habitat-forming species of one of the climax communities of some Mediterranean infralittoral areas due to the high productivity, complexity and stability of this vascular plant [1,2]. Meadows of this seagrass have an estimated coverage of 25–50% of infralittoral bottoms of the Mediterranean Sea (down to 50 m in certain areas), corresponding to the most extensive seagrass meadows for this basin [2,3]. These meadows support a highly diverse associated biotic community [1], providing an elevated number of ecosystem services such as the maintenance of marine biodiversity (including species of commercial interest), buffering of strong water movement leading to protection of the coastline against erosion, nutrient cycling and carbon sequestration, among others [2,4–7]. Despite all these services, *P. oceanica* has suffered a strong decline over the last centuries [7–9], with an estimated regression of meadows at 34% in the last 50 years as a consequence of the cumulative effects of multiple local stressors [10].

The decline and negative impacts suffered by this ecologically important species have led the EC to include *P. oceanica* meadows as priority habitat *Posidonia* beds (*Posidonion oceanicae*) (Habitat 1120) for conservation purposes within the EU Habitats Directive (92/43/EC). In addition, it is an indicator species and habitat for evaluating the "Good Environmental Status" of European waters in line with the Marine Strategy Framework Directive (MSFD, 2008/56/EC). Therefore, it is necessary to study the phenological parameters and the spatio-temporal variability of *P. oceanica* meadows across its biogeographical distribution, including the boundaries of its geographical distribution, as a preliminary step for further monitoring programs in order to improve the knowledge and management of this important marine ecosystem element.

*Posidonia oceanica* has its westernmost distribution limit close to the Strait of Gibraltar (SoG) in the northern Alboran Sea, being generally absent along the Moroccan coastline except for the Chafarinas Islands (35°05′ N, 02°25′ E) [10,11]. At its easternmost limit, *P. oceanica* occurs in a few localities along the southern Levantine part of Turkey close to the Kizilliman marine protected area and Alexandria (Egypt) on the southern coast [10,12,13]. In the Alboran Sea, *P. oceanica* forms extensive beds in its easternmost sector, close to the Almeria-Oran Front. In the central and western sectors, these beds are reduced from usually fragmented meadows to small patches [5,11]. Despite the fragmentation experienced by *P. oceanica* meadows close to the SoG, several authors have documented highly diverse associated invertebrate assemblages (e.g., molluscs [14] and crustacean decapods [15]). This fosters the idea that *P. oceanica* patches have an important ecological and conservation value whatever their size, as even small patches comprising a low number of shoots still support a higher abundance and diversity of invertebrates than adjacent sedimentary bottoms, as observed in other seagrass species [14,15].

The Alboran Sea displays transitional characteristics between the Atlantic Ocean and the Mediterranean Sea, with a predominance of colder and less salty water masses than other Mediterranean areas [16]. Moreover, the entrance of surface Atlantic waters through the SoG generates a series of anticyclonic gyres due to several physical factors (northeast orientation of the SoG, the Coriolis effect, the topography of the Alboran basin, the local climatology) that produce upwellings of cold and nutrient enriched Mediterranean deep waters in the northwestern Alboran coasts [17,18] (Figure 1). This sector of the Mediterranean Sea supports one of the highest productive areas within the region [19,20], which, together with the complex seafloor heterogeneity and the geographical location of the Alboran basin, promotes a high diversity of habitats and species [21,22], including ones of commercial importance. The latter has allowed the development of an intensive fishing activity targeting demersal, small pelagic and benthic species [23–26], with potentially adverse effects on vegetated benthic assemblages [27,28].

Meadows of *P. oceanica* have been widely studied in the Mediterranean Sea, with most studies being focused on seasonal and monthly variations [28–33], whereas few of them have considered interannual differences [34–36]. Despite the ecological importance of *P. oceanica*, studies on phenological aspects are scarce at the limits of the seagrass distribution range, including the Alboran Sea, where hydrological characteristics of the basin are very different from those in other parts of the Mediterranean Sea due to the Atlantic influence. Within this context, peripheral or edge populations are important for a species' long-term survival and evolution [37,38], as these populations usually present a high genetic differentiation compared to central populations due to reduced population size and genetic drift in fragmented habitats, which results in a lower genetic diversity [39]. Ecotypes derived from this genetic differentiation respond differently to environmental changes, being able to show even better tolerance to thermal shock than central populations [40]; however, their response depends on local population traits, as has been observed in some marine macrophytes (e.g., *Laminaria digitata*, *Fucus vesiculosus*, *Zostera marina*, *P. oceanica*) [40–42]. This highlights the importance of studying and conserving edge populations of marine macrophytes.

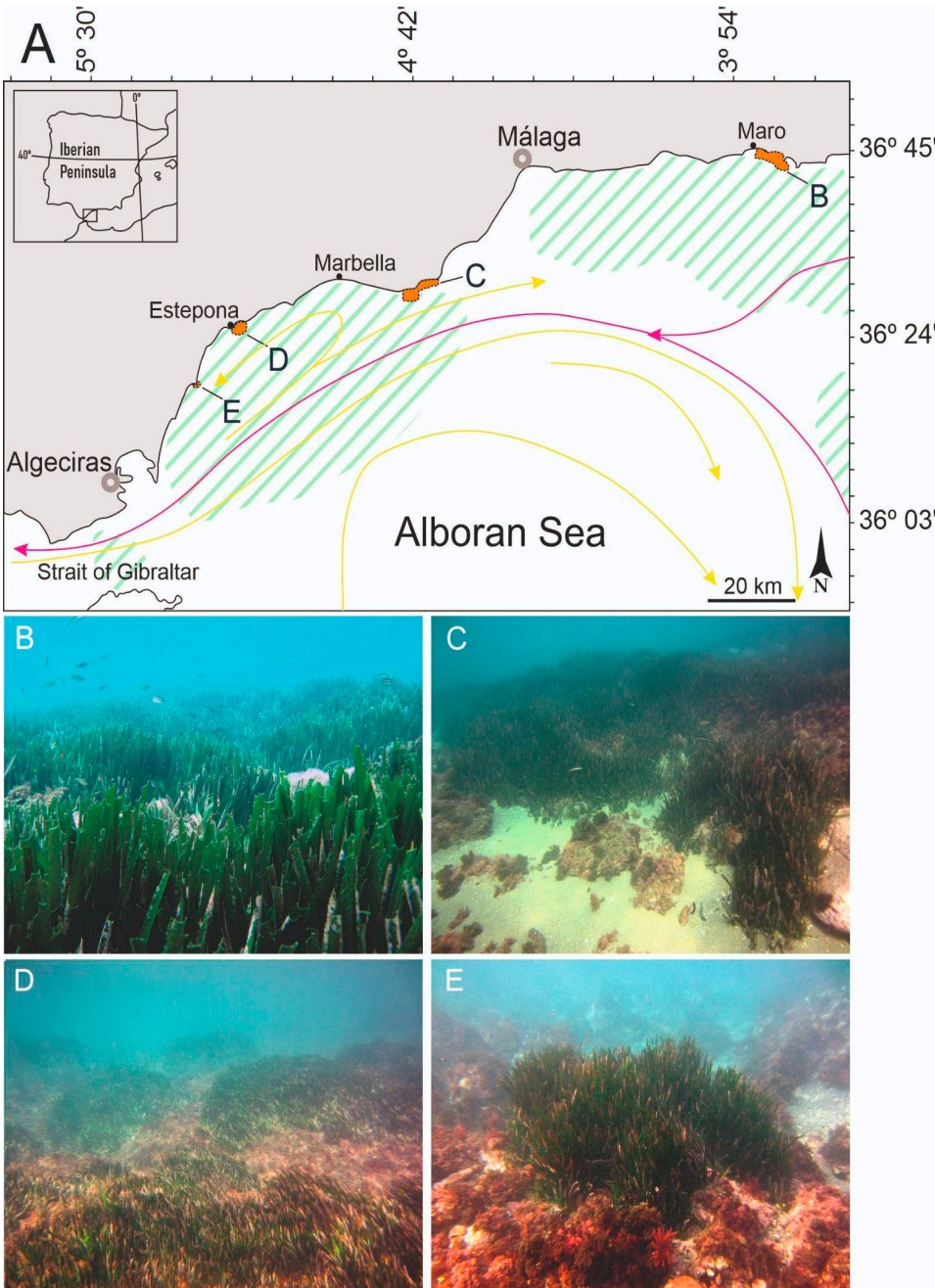

**Figure 1.** Map of the study area in the northwestern Alboran Sea and images of the general configuration of *Posidonia oceanica* patches in the studied meadows. (**A**) General map. Orange areas indicate the location of the different sites with *P. oceanica* meadows sampled: (**B**) Maro MPA, (**C**) Calahonda SAC, (**D**) Estepona SAC and (**E**) Chullera. Green lines indicate areas influenced by nutrient-rich upwellings; yellow arrows indicate the direction of the Surface Atlantic Waters and pink arrows indicate the direction of the Mediterranean Intermediate and Deep. Redrawn from http://www.juntadeandalucia.es/medioambiente/rediam (accessed on: 14 April 2022).

The main objectives of this study are (1) to characterize *P. oceanica* meadows at its westernmost gradient of distribution in the Mediterranean Sea; (2) to analyze the seasonal and interannual variation of different phenological parameters in the Mediterranean–Atlantic transition region in comparison to other Mediterranean areas; and (3) to evaluate flowering events in different *P. oceanica* meadows of the northwestern Alboran coasts.

## 2. Materials and Methods

### 2.1. Study Area

Characterization of phenological parameters of *P. oceanica* meadows, together with the assessment of flowering events, was carried out at four sites along the northwestern Alboran margin (Figure 1). These were, from east to west: the marine protected area "Paraje Natural Acantilados de Maro-Cerro Gordo" (hereafter Maro MPA) (Natura 2000 site code: ES6170002), the Special Area of Conservation "Calahonda" (hereafter Calahonda SAC) (Natura 2000 site code: ES6170030), the Special Area of Conservation "Fondos Marinos de la Bahía de Estepona" (hereafter Estepona SAC) (Natura 2000 site code: ES6170036) and Punta Chullera (hereafter Chullera) located just 40 km from the SoG.

Maro MPA is located between Nerja (Málaga) and Almuñécar (Granada) (Figure 1) and includes a narrow coastal strip (12 km) with a series of small beaches, soft and rocky bottoms, coves and cliffs, underwater seawalls and submerged caves [43]. This site covers 15.29 km$^2$ of the MPA with a depth range from the shore down to ca. 70 m. The studied *P. oceanica* meadow is located in a small cove with mixed pebble, rocky and sandy bottoms called "Molino de Papel" at a depth between 2 m (on rocky bottoms) and 12 m (on sandy bottoms). Calahonda SAC is located ca. 10 km further west between Fuengirola and Marbella, covering an area of 14 km$^2$ from the shore down to ca. 30 m depth (Figure 1). This site is characterized by the presence of different soft bottoms and rocky outcrops in the infralittoral zone, as well as intertidal sandy and rocky shores [43]. The studied meadows correspond to patches of variable sizes located on shallow rocky bottoms at 2–4 m depth. Further west, Estepona SAC is located between the harbour of Estepona and Punta de la Plata (Figure 1). This 6 km$^2$ MPA is separated by a distance of ca. 150 m from the shore and covers a depth range from 1 to ca. 50 m [43]. The studied meadow occurs on shallow rocky bottoms at depths not exceeding 4 m. Finally, the westernmost site was Chullera, a rocky cape with several coves and low cliffs located along the coast of Manilva and very close to the SoG and Atlantic Ocean (Figure 1). Soft bottoms predominate at Chullera, with rocky outcrops appearing in the shallowest area [43]. Here, *P. oceanica* occurs as very small patches on rocky bottoms at a depth of around 1–2 m. More information about these MPAs and on the general distribution maps of *P. oceanica* meadows can be found in Mateo-Ramírez et al. [43] and Ruiz et al. [44], respectively. Sea-surface temperature and salinity, although with temporal variations, generally increase from Chullera to Maro MPA due to the greater influence of Atlantic waters (less salty and warm) on the western sites, with values ranging from 16.9 °C to 18.3 °C and from 36.7 to 37.2 psu, respectively [45–47]. Wave action and tidal variation in the northwestern Alboran Sea are somehow influenced by its proximity to the Atlantic Ocean. For example, in Calahonda SAC, maximum wave height values are reported in autumn–winter (1–1.5 m) and, due to its proximity to the SoG, the frequent westerly and easterly winds (velocities ranging between 3.6–5.7 m·s$^{-1}$) favour constant wave action during the whole year (data taken from http://www.puertos.es accessed on: 15 February 2022 [48]).

### 2.2. Data Collection

The general characterization of the selected *P. oceanica* meadows was done using SCUBA diving during different years from 2006 to 2010 and 2015 when information about bathymetric range, substrate types where *P. oceanica* occurs, patch size, coverage, shoot density and other phenological parameters (e.g., leaf height, number of leaves per shoot, number of inflorescences per square meter), as well as environmental variables was collected.

The line intercept method was used for cover estimates and included the use of eight 25 m transects in each sampling station. Cover estimates corresponded to the percentage of the substrate covered by live *P. oceanica* in relation to the whole surface area indicated as "% coverage". On the other hand, a non-destructive technique was used for collecting shoot density and phenological data among the studied meadows within the 2–7 m depth range (depth of maximum coverage in most of the sites) from mid-November to mid-December of 2015. At each station, patches of different sizes were sampled, including three

small patches (patch size <1 m$^2$), three medium patches (1–5 m$^2$) and three large patches (>5 m$^2$). Three quadrats of 50 × 50 cm were deployed for shoot density measurements (number of shoots per m$^2$) in each of these patches (total = 27 quadrats per station). In the case of small patches, quadrats were deployed in directly adjacent patches as the patch size was too small for three replicates. The sampling area of the quadrats is similar to that used by other authors in previous studies on shallow Mediterranean seagrass beds [31,33,49,50] and is also used for seagrass monitoring [51]. In addition, 10 haphazardly selected shoots were analyzed in situ in each quadrat for leaf parameters (total = 270 shoots per station) following the methodology suggested by Gobert [35], such as the number of leaves per shoot (considering adult, intermediate and juvenile leaves) and the height (based on leaf length) of the highest (longest) leaf, which was measured to the nearest mm from the basal part of the sheath to the blade tip using a meter rule. Finally, comparative flowering data (number of flowers in the quadrats, i.e., flowering density) of *P. oceanica* was recorded across the northwestern Alboran Sea (Maro MPA, Calahonda SAC, Estepona SAC and Chullera) in November–December 2015. Flowering frequency was calculated as the percentage of flowering shoots in relation to the total number of shoots within the quadrats, according to Pergent et al. [30].

Seasonal and interannual variation of phenological parameters of *P. oceanica* was studied in greater detail at Calahonda SAC as it represents a site with a good representation of meadows located in an intermediate area within the seagrass westernmost geographical boundary. Moreover, this site is still highly influenced by surface Atlantic waters, which promotes the occurrence of a large number of algae species with Atlantic affinity [52,53] (Figure 1). Phenological and environmental data were recorded at two sampling stations within this site, "Punta de Calaburras" (hereafter Calaburras; 36°30.4′ N–04°38.3′ W) and "Calahonda" (36°29.2′ N–04°42.04′ W), the latter located 7 km to the west and ca. 100 km to the SoG. Shoot density and leaf parameters (the same mentioned above) were recorded seasonally from July 2007 to March 2010 in patches with sizes ranging between 1–130 m$^2$ at a depth of 2–3 m. Shoot density was estimated from 5 quadrats (50 × 50 cm) at each sampling station.

Assessment of leaf parameters at Calahonda SAC included quantification of the number of leaves per shoot counted in 10 shoots randomly selected in situ within each quadrat, together with measurements of the highest leaf height in 50 shoots per season and sampling station. When inflorescences were observed, flowering data (flowering density and frequency) were collected for studying the seasonal and interannual variation of flowering events. Additionally, the area of those patches containing inflorescences was measured.

In order to explore the relationships of seasonal and interannual changes of seagrass phenological parameters with those of environmental parameters, surface seawater temperature (SST), solar irradiance and concentration of chlorophyll *a* (Chl *a*) were measured 1–2 weeks before, during and after plant phenology data collection at Calaburras and Calahonda. SST was measured at midday (12:00) with an alcohol thermometer. Two replicates of 1 litre of surface seawater were collected per day at the site and transported in darkness at low temperatures to the laboratory for Chl *a* determination. Pigment analyses were carried out by filtering through Whatman GF/C glass filters, and extraction made using 100% acetone for 12h in cool and dark conditions. Measurements were made using a spectrophotometer at wavelengths of 630, 647, 664 and 750 nm. Concentrations of Chl *a* were obtained using the equation proposed by Jeffrey and Humphrey [54]. Solar irradiance measurements were obtained using HOBO Data Loggers (model UA-002-08, Onset) that were deployed seasonally from winter 2009 to winter 2010 (an annual cycle). Each HOBO was placed daily close to the studied seagrass meadows at 4–5 m depth and set to record data with 10 min intervals between 10:00 and 14:00. Additionally, the organic matter content (% OM) in sediment was also included in the analyses and measured from 5 replicates of sediment samples collected within the studied *P. oceanica* meadows or adjacent to patches

in each season. The % OM in samples of dry sediment (80 g) was obtained from the weight loss after ignition at 560 °C for 1 h.

Finally, the biotic index based on *P. oceanica* (BiPo) [55] was calculated in order to evaluate the conservation status of the different *P. oceanica* meadows studied and documented here. The BiPo index was developed to evaluate the ecological status of this seagrass, and it is in accordance with the EU Water Framework Directive requirements. The quantification of the ecological status is made according to the degree of deviation from reference conditions and expressed as a numerical value ranging from 1 (best conditions) to 0 (worst conditions), known as the ecological quality ratio (EQR) (see Lopez y Royo et al. [55] for more details).

### 2.3. Statistical Analyses

Differences in shoot density and phenological parameters (number of leaves, leaf height) were analyzed using a two-way PERMANOVA design 1 with two fixed factors: site (4 levels: Maro MPA, Calahonda SAC, Estepona SAC, Chullera); and patch size (3 levels: small, medium, large). Seasonal and interannual differences in *P. oceanica* dynamics were analyzed using a two-way PERMANOVA design 2 with two fixed factors: seasons (4 levels: Summer [July–September], Autumn [November–December], Winter [January–March], Spring [May–June]); and years (3 levels: A1 [July 2007–May 2008], A2 [September 2008–June 2009], A3 [September 2009–March 2010]). The two-way PERMANOVA design 2 was also used to test for statistical differences in environmental variables between seasons and years. Differences in inflorescence density could only be studied in Calahonda SAC meadows using a one-way PERMANOVA design 3 with one fixed factor: patch size (3 levels: small, medium, large). Analyses were based on Euclidean distances [56], and the significance of *p* values was determined through 9999 permutations of raw data and residuals under a reduced model for one-way and two-way analysis, respectively. When the numbers of permutations were low, the Monte Carlo method was used to contrast the significance of *p* values. Two-way PERMANOVA pair-wise post hoc tests ($p < 0.05$) were used for posterior multiple comparisons across months and years in design 2. In order to know the magnitude of temporal fluctuations in shoot density and phenological parameters, the coefficient of variation (CV, %) was calculated with all collected data.

When seagrass parameters, sediment and environmental variables displayed a similar pattern, forward stepwise multiple linear regression analyses were used to investigate the corresponding relationships. Solar irradiance was analyzed independently because data was only available from March 2009 to March 2010. Prior to carrying out the regression analyses, potential significant relationships between the independent variables (seagrass parameters, sediment and water column variables) were investigated using Pearson's correlation coefficient, and those that were significantly and highly correlated were eliminated from the analyses. Data used in the regression analyses were checked for normality using Kolmogorov and Smirnov test and transformed to natural logarithms when necessary. These statistical procedures were performed using the software SPSS 20 and PRIMER 6.0, and PERMANOVA statistical package.

## 3. Results

### 3.1. Spatial Variability along the Northwestern Alboran Sea

The bathymetric distribution of *P. oceanica* meadows within each site displayed a decreasing trend towards the Strait of Gibraltar (SoG), with those of the Maro MPA (easternmost station) displaying a wide depth range (2–12 m depth); those of Calahonda SAC displaying an intermediate depth range (1–5 m depth); and those of Estepona SAC and Chullera (westernmost stations) displaying a very narrow depth range (1–2.5 m and 1–1.5 m depth, respectively).

The studied *P. oceanica* meadows were generally fringing or growing on rocky outcrops, displaying a gradient in the meadow structure from continuous meadows to small patches

towards the SoG. The meadows at Maro MPA presented small patches (0.25–0.5 m$^2$) at shallow depths (2–3 m depth) and medium to large patches (up to 8 m$^2$) at a depth of 6 m depth, where it generally changed to a more or less continuous bed (Figure 1). Meadows of *P. oceanica* at Calahonda SAC and Estepona SAC were structured as medium to large patches, sometimes becoming fragmented meadows (Figure 1). Finally, the meadows at Chullera were reduced to a few small to medium patches (0.5–1.5 m$^2$) located on very shallow rocky outcrops (Figure 1).

Coverage of *P. oceanica* within the 1 to 6 m depth range displayed a decreasing trend towards the SoG, ranging from ca. 50% in Maro MPA to ca. 45% in Calahonda SAC and Estepona SAC and less than 5% in Chullera with a low number of patches of small size. In Maro MPA, *P. oceanica* coverage also decreased with depth, showing values lower than 10% at its lower limit (12 m depth). Dead matte heavily covered by macroalgae (mainly *Halopteris scoparia*) was detected between patches of living matte in most study sites (Figure 2), especially at Calahonda SAC, resulting in isolated and/or fragmented seagrass populations.

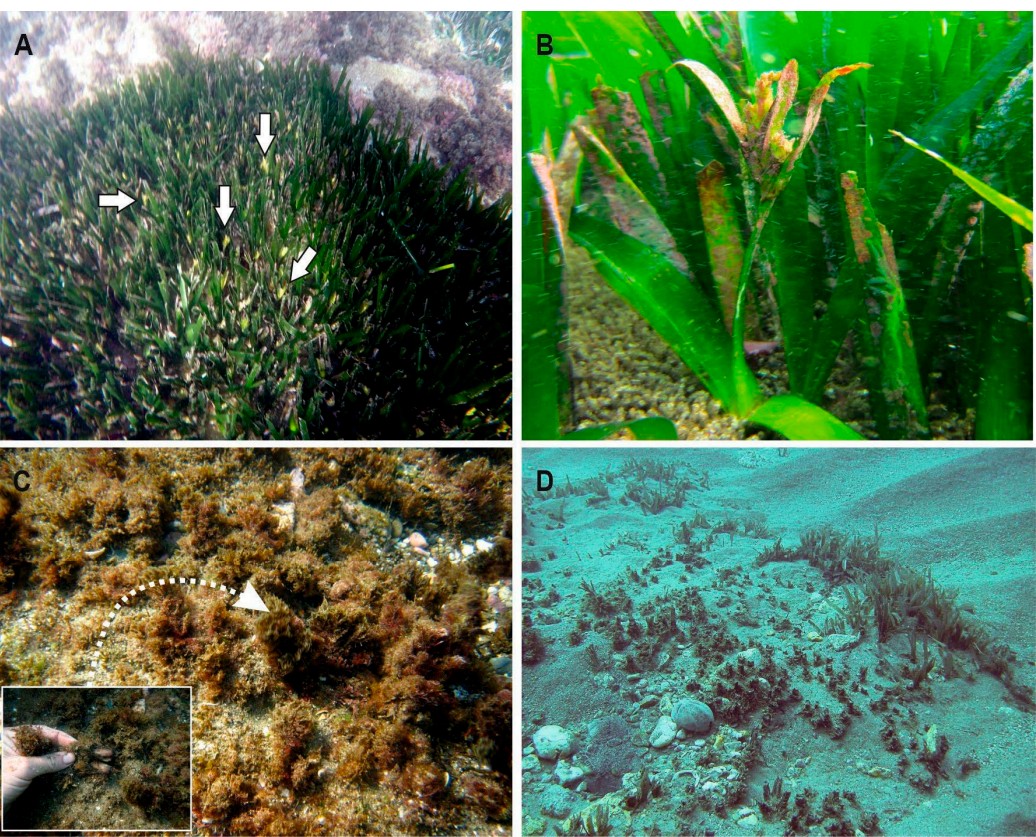

**Figure 2.** (**A**) *Posidonia oceanica* patch with several inflorescences (arrows) and (**B**) detail of one inflorescence from Calahonda SAC meadows, November 2015. (**C**) Dead matte covered by macroalgae (mainly *Halopteris scoparia*) from Calahonda SAC meadows. (**D**) Dead matte from Maro MPA meadows after burial due to sediment transport linked to a torrential rainfall in September 2007.

An increasing trend of shoot density was detected towards the SoG, with the largest shoot density mean values detected at Chullera (753 ± 63 shoots m$^{-2}$; mean ± SE) and Estepona SAC (662 ± 27 shoots m$^{-2}$), whereas similar values were reported at Maro MPA (550 ± 44 shoots m$^{-2}$) and Calahonda SAC (514 ± 42 shoots m$^{-2}$) (Figure 3A). Mean maximum leaf height was highest at Chullera (25.8 ± 3.8 cm) and Estepona SCI (25.3 ± 1.2 cm), followed by Calahonda SAC (22.1 ± 1.4 cm) (Figure 3B; Table 1) and Maro MPA (17.6 ± 0.9 cm) (Figure 3B). The mean number of leaves per shoot showed a significant decreasing trend from Maro MPA (6.6 ± 0.3 leaves shoot$^{-1}$) to Chullera (5.9 ± 0.4 leaves shoot$^{-1}$), with intermediate values

at Calahonda SAC and Estepona SAC ($6.1 \pm 0.2$ and $6.3 \pm 0.3$ leaves shoot$^{-1}$), respectively (Figure 3C; Table 1).

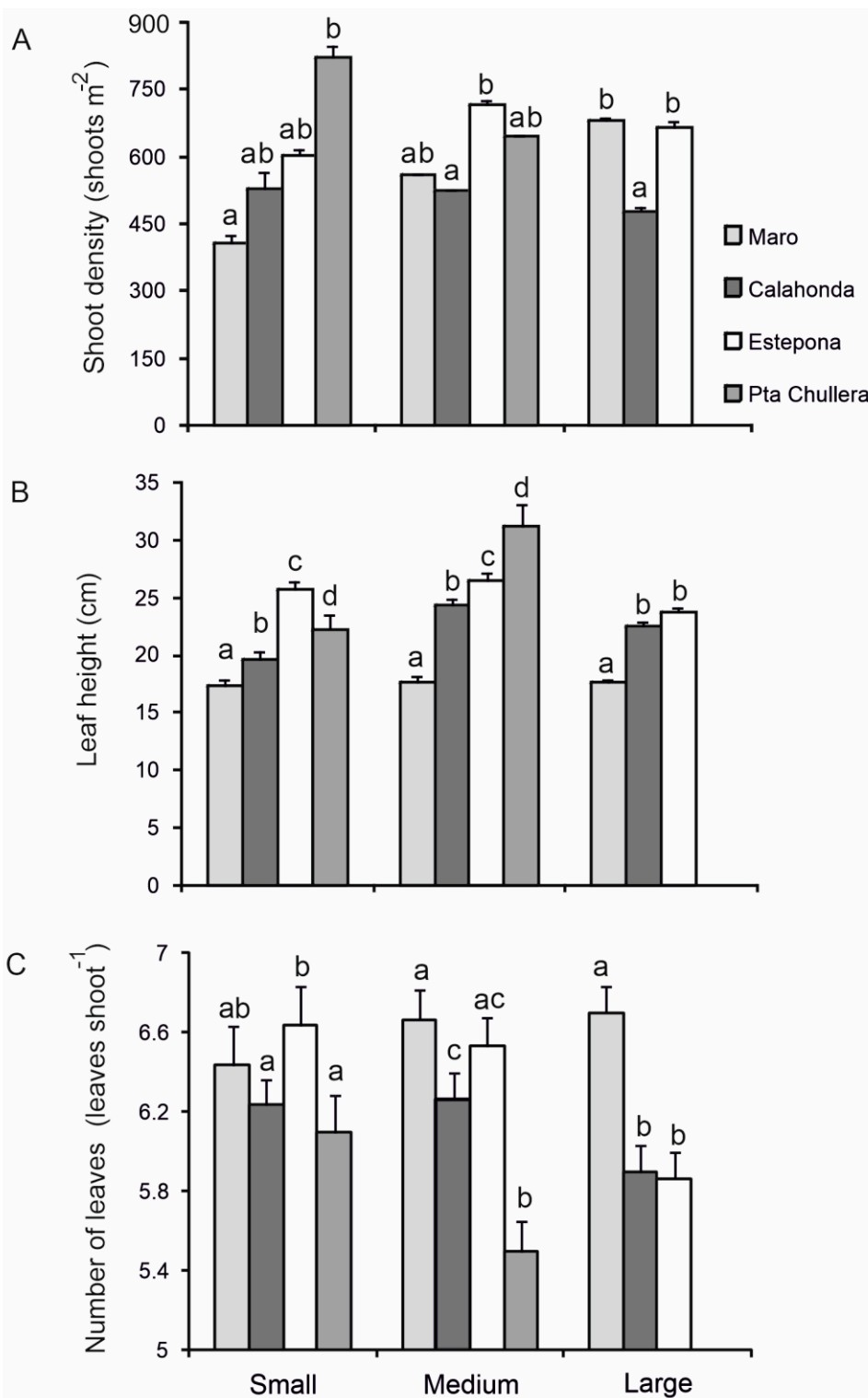

**Figure 3.** Shoot density and phenological parameters of *Posidonia oceanica* as a function of patch sizes and sites. (**A**) Shoot density (shoots m$^{-2}$), (**B**) maximum leaf height (cm) and (**C**) number of leaves per shoot (leaves shoot$^{-1}$). Different letters indicate significant differences within each patch size category ($p < 0.05$). Mean + standard error.

**Table 1.** Results of two-way PERMANOVA test for differences in shoot density, maximum leaf height, and number of leaves of *Posidonia oceanica* amongst sites and patch size. Results of one-way PERMANOVA test for differences in inflorescence density of *P. oceanica* meadows of Calahonda SAC amongst patch sizes. Significant differences are indicated by: ** $p < 0.01$; *** $p < 0.001$.

| Source of Variation | df | MS | F | p |
|---|---|---|---|---|
| Shoot density | | | | |
| Site | 3 | 77,026 | 5.7217 | ** |
| Patch size | 2 | 9290.4 | 0.69011 | 0.5228 |
| Site × Patch size | 5 | 30,653 | 2.2769 | 0.0767 |
| Residual | 21 | 13,462 | | |
| Total | 31 | | | |
| Leaf height | | | | |
| Site | 3 | 1215.5 | 66.207 | *** |
| Patch size | 2 | 397.53 | 21.653 | *** |
| Site × Patch size | 5 | 143.97 | 7.8415 | *** |
| Residual | 309 | 18.359 | | |
| Total | 319 | | | |
| Number of leaves | | | | |
| Site | 3 | 8.7974 | 13.739 | *** |
| Patch size | 2 | 3.1323 | 4.8916 | ** |
| Site × Patch size | 5 | 2.4972 | 38,998 | ** |
| Residual | 309 | 0.64035 | | |
| Total | 319 | | | |
| Inflorescences density | | | | |
| Patch size | 2 | 15,900 | 2.1032 | 0.2107 |
| Residual | 6 | 7559.8 | | |
| Total | 8 | | | |

Regarding patch size, and despite non-significant differences detected in shoot density with patch size (Table 1), the pairwise analysis showed statistical differences for the interaction term of site and patch size (Figure 3A). The highest shoot density values were recorded in small-size patches of Chullera ($823 \pm 27$ shoots m$^{-2}$), whereas the lowest values were recorded in small-size patches of Maro MPA ($406 \pm 20$ shoots m$^{-2}$) (Figure 3A). On the other hand, the highest leaf height values observed at the time of the study (November–December) were generally detected in medium-size patches at all sites, except those of Maro MPA, with the highest mean values observed at Chullera ($31.2 \pm 1.9$ cm). Meadows at Maro MPA showed similar leaf height independent of patch size and had the lowest value from all the studied sites ($17.7 \pm 1.9$ cm) (Figure 3B; Table 1). Finally, the number of leaves per shoot was significantly higher in small-size and medium-size patches, ranging between $6.7 \pm 0.7$ and $6.1 \pm 1$ leaves shoot$^{-1}$ (Figure 3C; Table 1).

*3.2. Flowering*

Three flowering events were detected in the *P. oceanica* meadows of Calahonda SAC during the six years of monitoring (2006–2010 and 2015). Mature inflorescences were observed at Calaburras in March 2009 on orthotropic shoots of several patches with areas ranging between 4 and 10 m$^2$ at a depth of ca. 2 m, with a mean value of $12.7 \pm 4.1$ inflorescences per m$^2$ (infl. m$^{-2}$) that represented a 1.9% flowering frequency (flow. freq.), with a maximum density of 28 infl. m$^{-2}$. A second and major flowering event was observed in November 2009 in almost all patches studied in Calahonda SAC, with mean values of $34.0 \pm 7.4$ infl. m$^{-2}$ at Calaburras ($7.0 \pm 3.0$% flow. freq.) and of $69.0 \pm 15.3$ infl. m$^{-2}$ at Calahonda ($6.7 \pm 3.4$% flow. freq.), and with maximum densities of 144 and 184 infl. m$^{-2}$, respectively. Finally, a massive flowering event was detected at Calahonda in November 2015, with a mean value of $115.2 \pm 98.2$ infl. m$^{-2}$ ($22.3 \pm 6.6$% flow. freq.) and a maximum density of 330 infl. m$^{-2}$ (Figure 2A). Here, PERMANOVA analyses did not reveal significant differences in inflorescence density among patch sizes (Table 1). On this last occasion, no inflorescences were detected at Maro MPA, Estepona SAC and Chullera.

### 3.3. Seasonal and Interannual Variability

Mean surface seawater temperatures (SST) showed a clear temporal trend at both Calahonda and Calaburras sampling stations, with values ranging from 22 °C in warm months (with exceptional values of 27 °C in September 2009) to 15 °C in cold months (Figure 4A). Seasonal differences in SST were significant at both sampling stations, whereas annual differences were only significant at Calaburras (Table 2). The Chl *a* concentration displayed a similar temporal trend, with the highest values in warm months and the lowest ones in cold months, with Calahonda generally displaying higher values (Figure 4B). Regarding Chl *a*, seasonal differences were significant at both sampling stations, with significant annual differences only at Calahonda (Table 2). Solar irradiance displayed a temporal trend with high values in warm months and low ones in cold months (Figure 4C), but no significant seasonal differences were detected (Table 2). The percentage of organic matter in sediments collected within the seagrass patches showed a similar seasonal trend at both sampling stations, with high values in warm months (ca. 2.4%) and low ones in cold months (ca. 1.6%) (Table 2).

**Table 2.** Results of one and two-way PERMANOVA test for spatial and temporal changes of environmental variables in *Posidonia oceanica* meadows at Calaburras and Calahonda (Calahonda SAC, northwestern Alboran Sea) in relation to sampling stations, months from different seasons and years (annual cycles). Significant differences are indicated by: * $p < 0.05$; ** $p < 0.01$; *** $p < 0.001$.

| Source of Variation | | Surface Seawater Temperature | Solar Irradiance | Chlorophyll *a* | Percentage of Organic Matter |
|---|---|---|---|---|---|
| Calaburras | Year | F = 4.033 * | – | F = 2.896 $p = 0.068$ | F = 0.898 $p = 0.414$ |
| | Season | F = 75.677 *** | F = 2.075 $p = 0.164$ | F = 5.324 ** | F = 6.474 *** |
| | Year × Season | F = 19.292 *** | – | F = 4.971 ** | F = 1.315 $p = 0.263$ |
| Calahonda | Year | F = 1.172 $p = 0.344$ | – | F = 6.575 ** | F = 1.822 $p = 0.171$ |
| | Season | F = 8.872 ** | F = 1.410 $p = 0.307$ | F = 4.410 * | F = 10.323 *** |
| | Year × Season | F = 0.646 $p = 0.639$ | – | F = 3.947 * | F = 2.324 * |
| Among sampling stations | | F = 0.195 $p = 0.657$ | F = 3.410 $p = 0.078$ | F = 7.136 ** | F = 4.498 * |

The mean shoot density of *P. oceanica* was not significantly different amongst years (annual cycles) and sampling stations of Calahonda SAC (one-way PERMANOVA; F = 0.031, $p = 0.862$), with very similar values of 850 $\pm$ 15 shoots m$^{-2}$ at Calaburras and 845 $\pm$ 20 shoots m$^{-2}$ at Calahonda (Table 3). Seasonal (monthly) variability was only significant at Calahonda, where shoot density displayed a temporal trend with higher values in cold months (maximum: 1055 shoots m$^{-2}$, November 2007) during the first two years (Figure 5A; Table 3), whereas at Calaburras shoot density was very similar throughout the year (Figure 5A; Table 3, Supplementary Table S1). Finally, shoot density displayed negative correlations with the number of leaves ($R_{Pearson} = -0.387$, $p < 0.01$), as well as with maximum leaf height ($R_{Pearson} = -0.268$, $p < 0.01$).

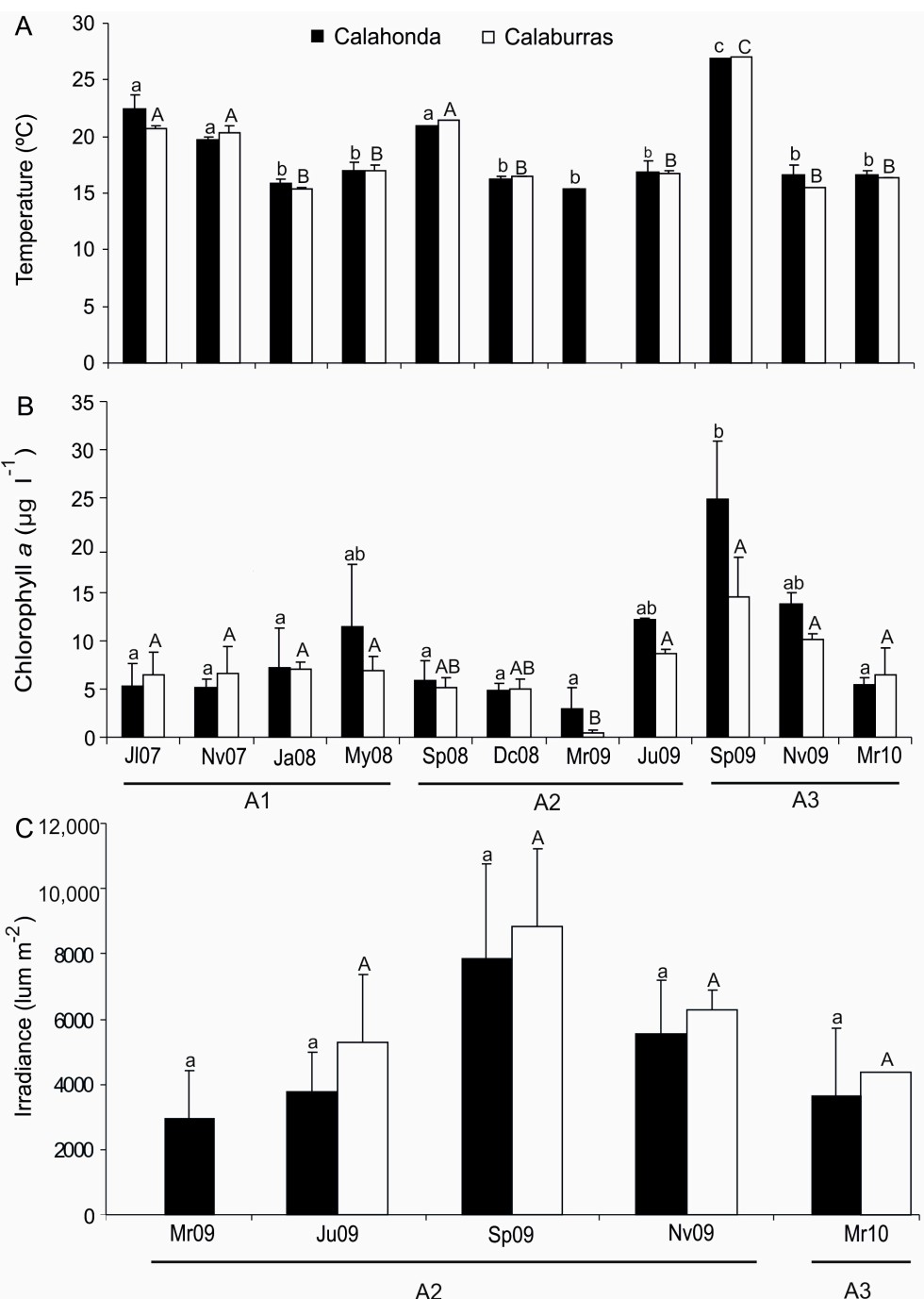

**Figure 4.** Temporal changes of environmental variables in *Posidonia oceanica* meadows in Calaburras (empty bars) and Calahonda (solid bars) sampling stations within the Calahonda SAC in different months/seasons throughout the first (A1), second (A2) and third (A3) annual cycles: (**A**) Surface seawater temperature, (**B**) concentration of Chlorophyll *a* and (**C**) solar irradiance in different seasons. July, Jl; November, Nv; January, Ja; May, My; September, Sp; December, Dc; March, Mr; June, Ju. Different letters indicate significant differences between months, with uppercase letters for Calaburras and lowercase letters for Calahonda ($p < 0.05$). Mean + standard error.

**Table 3.** Results of two-way PERMANOVA tests for differences in the shoot density, maximum leaf height and number of leaves per shoot of *Posidonia oceanica* between seasons and years. Significant differences * $p < 0.05$; *** $p < 0.001$.

| Source of Variation | | Calaburras | | | | Calahonda | | |
|---|---|---|---|---|---|---|---|---|
| | df | MS | F | *p* | df | MS | F | *p* |
| Shoot density | | | | | | | | |
| Year | 2 | 12,801 | 1.178 | 0.324 | 2 | 14,726 | 1.616 | 0.206 |
| Season | 3 | 11,404 | 1.05 | 0.382 | 3 | 162,810 | 17.867 | *** |
| Year × Season | 5 | 23,106 | 2.127 | 0.080 | 5 | 50,290 | 5.519 | *** |
| Residual | 43 | 10,865 | | | 43 | 9112.4 | | |
| Total | 53 | | | | 53 | | | |
| Leaf height | | | | | | | | |
| Year | 2 | 136.59 | 8.005 | * | 2 | 214.8 | 20.952 | *** |
| Season | 3 | 1449 | 84.921 | *** | 3 | 1052.8 | 102.7 | *** |
| Year × Season | 5 | 97.508 | 5.715 | *** | 5 | 32.055 | 3.1267 | * |
| Residual | 43 | 17.062 | | | 43 | 10.252 | | |
| Total | 53 | | | | 53 | | | |
| Number of leaves | | | | | | | | |
| Year | 2 | 0.361 | 2.583 | 0.084 | 2 | 0.098 | 0.751 | 0.473 |
| Season | 3 | 1.39 | 9.943 | *** | 3 | 1.713 | 13.176 | *** |
| Year × Season | 5 | 0.232 | 1.663 | 0.167 | 5 | 0.175 | 1.349 | 0.267 |
| Residual | 43 | 0.14 | | | 43 | 0.13 | | |
| Total | 53 | | | | 53 | | | |

Maximum leaf height displayed similar values in both sampling stations, with mean values ranging between $26.6 \pm 1.4$ cm at Calahonda and $24.9 \pm 1.2$ cm at Calaburras (one-way PERMANOVA: F = 0.863, $p = 0.357$). A temporal trend with significantly higher leaf height in cold months (maximum 39 cm in both sites, May 2008) was observed in both sampling stations (Figure 5B; Table 3; Supplementary Table S1), as well as significant interannual differences with mean annual values ranging between $22.5 \pm 1.7$ and $27.6 \pm 2.5$ cm (Figure 5B; Table 3; Supplementary Table S1).

The mean number of leaves per shoot was similar at both sampling stations (Calaburras: $5.7 \pm 0.1$ leaves shoot$^{-1}$; Calahonda: $5.6 \pm 0.1$ leaves shoot$^{-1}$) (one-way PERMANOVA; F = 0.422, $p = 0.519$), with no significant differences amongst annual cycles. On the other hand, the number of leaves displayed a temporal trend with significant seasonal (monthly) differences and maximum values in cold months (maxima: $6.3 \pm 0.1$ leaves shoot$^{-1}$ at Calahonda, March 2009; $6.1 \pm 0.1$ leaves shoot$^{-1}$ at Calaburras, January 2008) (Figure 5C; Table 3; Supplementary Table S1).

### 3.4. Relationships between Phenological, Sediment and Environmental Variables

This analysis could be performed in the seasonal and intra-annual study carried out at Calahonda SAC. Here, results for the irradiance showed the highest values of R squared and explained 47.9% and 59.4% of the variability in relation to the number of leaves and shoot density, respectively (Table 4).

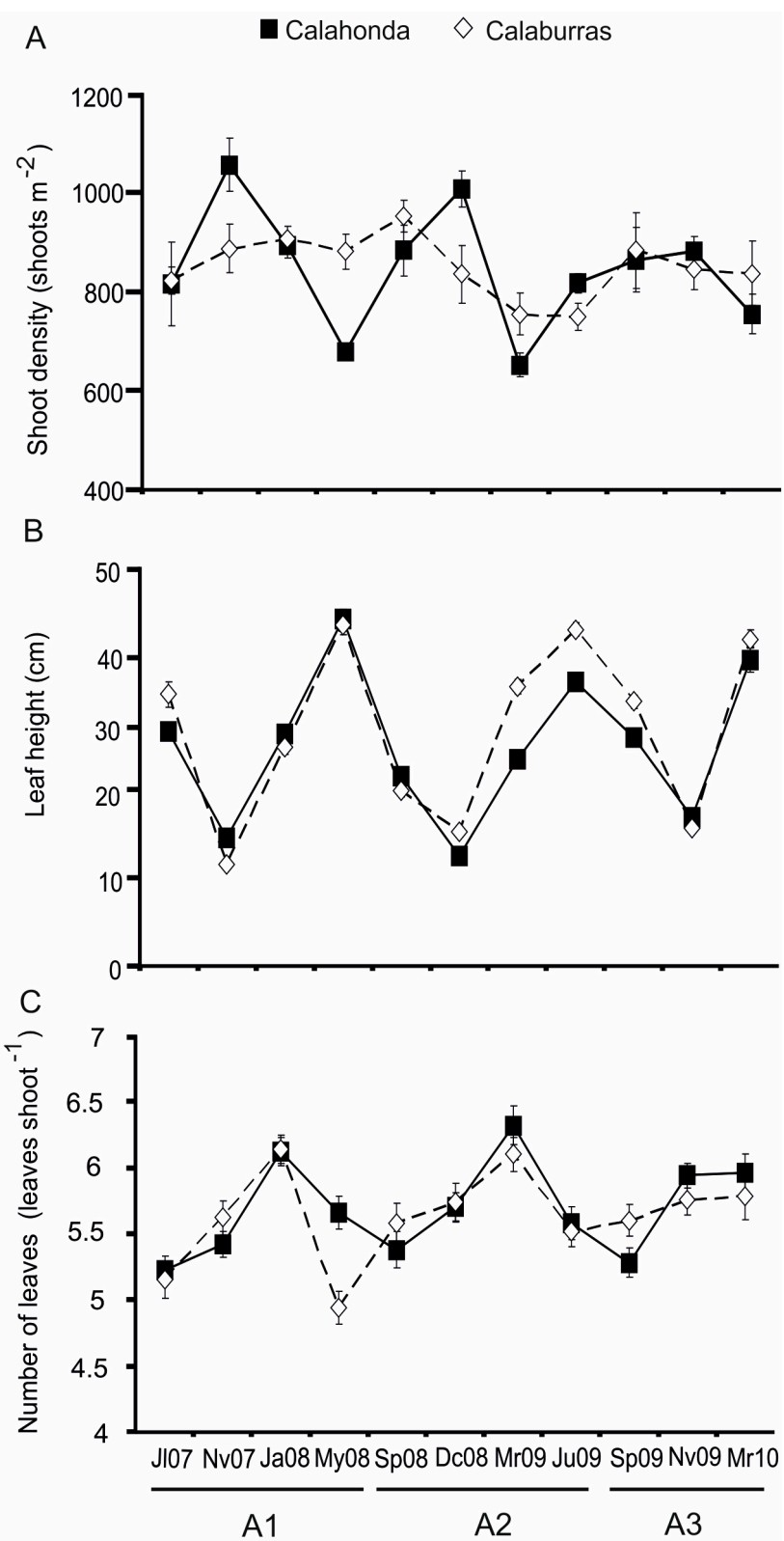

**Figure 5.** Temporal changes in shoot density, leaf height and number of leaves of *Posidonia oceanica* meadows at Calaburras empty rhombus and Calahonda solid square within the Calahonda SAC, in different months/seasons throughout the first (A1), second (A2) and third (A3) annual cycles: (**A**) Shoot density (shoots m$^{-2}$), (**B**) maximum leaf height (cm) and (**C**) number of leaves shoot$^{-1}$). July, Jl; November, Nv; January, Ja; May, My; September, Sp; December, Dc; March, Mr; June, Ju. Mean ± standard error.

**Table 4.** Results of regression analyses of *Posidonia oceanica* for shoot density and number of leaves with irradiance (for only one annual cycle) and surface seawater temperature. Only environmental variables accepted in each regression model ($p < 0.05$) are listed. * $p < 0.05$.

| | Coefficient | SE | F Ratio | $R^2$ | $p$ |
|---|---|---|---|---|---|
| Number of leaves | | | | | |
| Constant | 6.308 | 0.311 | | | |
| Temperature | −0.037 | 0.17 | 4.573 | | |
| Number of leaves | | | | 0.479 | * |
| Constant | 10.635 | 1.929 | | | |
| Irradiance | −1.319 | 0.52 | 6.424 | | |
| Shoot density | | | | 0.594 | * |
| Constant | −571.8 | 431.4 | | | |
| Irradiance | 372.4 | 116.3 | 10.255 | | |

*3.5. Status*

Most of the studied meadows presented a good conservation status during the sampling period (2007–2015), as reflected in the EQR value of the BiPo index ranging between 0.57 and 0.64 (Table 5). The exception was the meadow studied at Maro MPA, where a lower EQR value indicates a moderate conservation status (Table 5). The majority of meadows presented a sharp lower limit, which was located on shallow bottoms (5 m depth or less), and elevated values of shoot density and leaf height (Table 5). In the case of Maro MPA, meadows showed a sparse and deeper lower limit with lower values of shoot density and leaf height (Table 5).

**Table 5.** Evaluation of the different *Posidonia oceanica* meadows studied in the northwestern Alboran Sea according to the BiPo index (Biotic Index *Posidonia oceanica*) [55]. Ecological quality ratio (EQR) of the BiPo index is indicated, with values ranging between 1 (in reference condition) and 0 (in the worst condition) according to [55].

| Site-Sampling Station (Year) | | Lower Limit Depth (m) | Lower Limit Type | Shoot Density (Shoots $m^{-2}$) | Leaf Height (mm) | EQR | Class |
|---|---|---|---|---|---|---|---|
| Chullera | (2015) | 2 | Sharp | 753 | 258 | 0.63 | Good |
| Estepona SAC | (2015) | 4 | Sharp | 662 | 253 | 0.61 | Good |
| Calahonda SAC | Calaburrras (2007–2010) | 5 | Sharp | 849 | 250 | 0.61 | Good |
| | Calahonda (2007–2010) | 5 | Sharp | 945 | 237 | 0.64 | Good |
| | Calahonda (2015) | 5 | Sharp | 514 | 221 | 0.57 | Good |
| Maro MPA | (2015) | 12 | Sparse | 550 | 176 | 0.52 | Moderate |

**4. Discussion**

The distribution of *Posidonia oceanica* along the northern Alboran Sea, where its western limit of distribution is located, is not uniform. Here, the most extensive and continuous beds occur off El Ejido (Almería) at the Site of Community Importance, "Fondos Marinos de Punta Entinas-Sabinar" (code ES6110009) [11]. From here, towards the Strait of Gibraltar (SoG) and the Atlantic Ocean, *P. oceanica* meadows are becoming increasingly fragmented, and their extent is considerably reduced. The *P. oceanica* meadows studied along the northwestern Alboran Sea display a decreasing trend of patch size, bathymetric range and the number of leaves per shoot on moving toward the SoG, as well as an increasing trend of shoot density and leaf height. Similarly, the maximum depth reached by *P. oceanica* meadows decreases westwards from Punta Entinas-Sabinar (ca. 15 m depth) [57] to Chullera (1–1.5 m depth). The influence of surface Atlantic water entering eastwards through the SoG, together with the characteristic oceanographic and hydrological conditions of the northern Alboran Sea (e.g., higher turbidity, stronger wave action) [5,44,58], might represent

a developmental limiting factor for *P. oceanica* in this part of the Mediterranean. In this respect, other studies have documented the presence of other habitat types at unusually shallow bottoms in the northwestern Alboran Sea, such as coralligenous assemblages found at a depth of 15 m, which is usually characteristic of the circalittoral biocenosis [59,60]. This indicates lower water transparency in the northwestern Alboran Sea compared to other Mediterranean areas, as has also been detected in the adjacent Gulf of Cádiz. Moreover, the analyzed *P. oceanica* meadows are located in one of the highest biological productivity areas within the Mediterranean [61,62], which may also induce low water transparency. This was reflected in the low irradiance values obtained at a depth of 4–5 m at the Calahonda SAC, which are similar to those found at a depth of 10 m in the eastern Adriatic Sea [50].

Solar irradiance and seawater temperature are key factors modulating the abundance and productivity of seagrasses [8,12,63]. Indeed, light and surface seawater temperature explained a high percentage of variability of leaf number and shoot density of *P. oceanica* at the Calahonda SAC throughout the year. On the other hand, a narrower bathymetric distribution (ranging between 1–5 m depth) exposes the meadows to continuous wave action and turbulence, especially in the areas close to the SoG, where patches were very close to the surf zone (1–1.5 m depth). Regarding this aspect, shallow patchy meadows of different seagrasses have been reported to be affected by such extreme conditions (e.g., low water transparency and strong wave action), including *P. oceanica* meadows at its northern distributional boundary in the Adriatic Sea [64–67]. Within this context, studies carried out on *P. oceanica,* and other seagrasses show a negative relationship between seagrass biomass, leaf height, epiphyte biomass, leaf width and shoot area with wave energy [68,69]. Moreover, high-energy environments support *P. oceanica* meadows landscapes with higher patch numbers, lower coverage, more complex patch shapes and lower intra-patch architecture than sites exposed to lower wave action [69].

Shoot densities of the studied *P. oceanica* meadows were generally higher than those located in other Mediterranean areas at similar depths, including the north Adriatic Sea [66]) and the northeastern Levantine Sea (eastern distributional boundary [12]), but similar to those reported in meadows in Tunisia growing on a rocky substratum, as the ones studied here [70] (see Figure 6). Regarding leaf height, values recorded in the present study were lower than those documented in other Mediterranean areas (see Figure 6). Regarding this, Marbà et al. [8] reported *P. oceanica* meadows with high shoot densities and low leaf height values in other northern Alboran Sea locations [8]. Several physical factors acting together seem to highly influence the westernmost *P. oceanica* meadows, such as the shallow location and the rocky nature of the substratum, together with higher hydrodynamic exposure due to wave action. Furthermore, they may represent key drivers structuring ecological features and phenological parameters of these meadows along the northern Alboran Sea [11,68,69], being particularly evident at the westernmost meadows of Estepona SAC and especially of Chullera. This could represent a stress reduction strategy that involves increasing rhizome growth for anchorage as well as self-shading effects [35,71,72], as suggested by the negative correlations observed at Calahonda SAC between shoot densities, leaf height and the number of leaves per shoot.

The temporal trend observed for leaf height and number of leaves per shoot is similar to that reported for other *P. oceanica* meadows of both European and African coasts [30–32,36,73,74] Regarding shoot density and despite being a more stable descriptor, higher values were observed in cold months in Calahonda SAC. Moreover, an unusual seasonal trend was observed during the first two years, similar to that previously documented in other meadows, some of which were located close to the northern and eastern distribution edges of *P. oceanica* in Tunisia and Turkey [36,71,75]. Nevertheless, shoot density and number of leaves per shoot displayed lower intra-annual fluctuations when compared to other *P. oceanica* meadows (see Table 6), probably linked to the particular environmental characteristics of the northwestern Alboran Sea, including lower intra-annual temperature changes (15.3–22.5 °C) in comparison to other Mediterranean areas (e.g., Adriatic Sea: 9–24 °C; Tunisian coasts: 14–31 °C; southern Italian coasts: 11.8–24.8 °C [33,73,76]),

which may highly influence these meadows due to their shallow location. On the other hand, maximum leaf height displayed high differences amongst years, as observed previously [8,35,36]. Interannual studies of phenological parameters are scarce, but all of them have indicated some interannual variability, highlighting the need for performing long-term studies of meadow population characters and phenological parameters [8,35,36].

**Table 6.** Magnitude of intra-annual fluctuations (coefficient of variation, %) in shoot density and phenological parameters of *Posidonia oceanica* meadows in Calaburras and Calahonda (NW Alboran Sea), as well as in other locations of the Mediterranean Sea.

| Studied Area | Country | Reference | Shoot Density | Leaf Height | Number of Leaves |
|---|---|---|---|---|---|
| Calaburras | Spain | Present study | 13.1 | 38.7 | 8.5 |
| Calahonda | Spain | Present study | 17.4 | 34.7 | 8.3 |
| Tabarca | Spain | [31] | | 58.4 | 10.7 |
| Banyuls-sur-Mer | France | [29] | | 23.5 | 6.7 |
| Port-Cros | France | [29] | | 39.6 | 19.4 |
| La Revellata Bay (Corsica) | France | [34] | 20.6 | 30 | 18.4 |
| Monterosso al Mare | Italy | [35] | 42.4 | 43.7 | 20.6 |
| Lacco Ameno (Ischia Island) | Italy | [1] | | 32.3 | 10.4 |
| Otranto (Apulia) | Italy | [32] | 41.2 | 34.5 | 3.6 |

The inflorescences followed by fruit development observed at Calahonda SAC would indicate that sexual reproduction of *P. oceanica* may potentially occur in meadows at their western limit of geographical distribution [80]. Flowering of *P. oceanica* is highly variable within and between meadows, with differences even at small spatial scales of a few meters [81]. Despite this, the flowering and recruitment of this Mediterranean seagrass seem to be more frequent than previously expected [82,83]. The available information seems to indicate that flowering represents an irregular event mainly related to high seawater temperature [84,85]. In fact, three flowering events were registered in the meadows at Calahonda SAC between 2006 and 2015, with the highest flowering episode coinciding with the warmest seawater temperature values ever recorded for the area at that time. Regarding this, it is remarkable that the high flowering frequency observed in 2015, with similar or even higher values than those reported in meadows of the central and eastern Mediterranean basin [75,86,87]. Nevertheless, no flowering events were detected in Maro MPA, Estepona SAC, or Chullera during that year. This is of interest as Calahonda SAC meadows are located close to the westernmost distributional limit of *P. oceanica*, which would indicate (1) that these meadows present an adequate health status for sexual reproduction; or (2) a survival mechanism in response to natural or anthropogenic disturbances [85,88]. Furthermore, the intensity of the flowering and the absence of significant differences amongst patches of different sizes in relation to inflorescence density may be related to the latter notion. Fruit production and seed germination still need to be monitored in these *P. oceanica* meadows in long-term studies in order to evaluate the extent of sexual reproduction in the formation of new patches. This is of importance to evaluate the survival capacity of *P. oceanica*, as edge populations could play an important role in the long-term survival of this seagrass; in fact, a recent study shows that thermal edge populations of *P. oceanica* have a greater resilience against thermal stress than central populations [42].

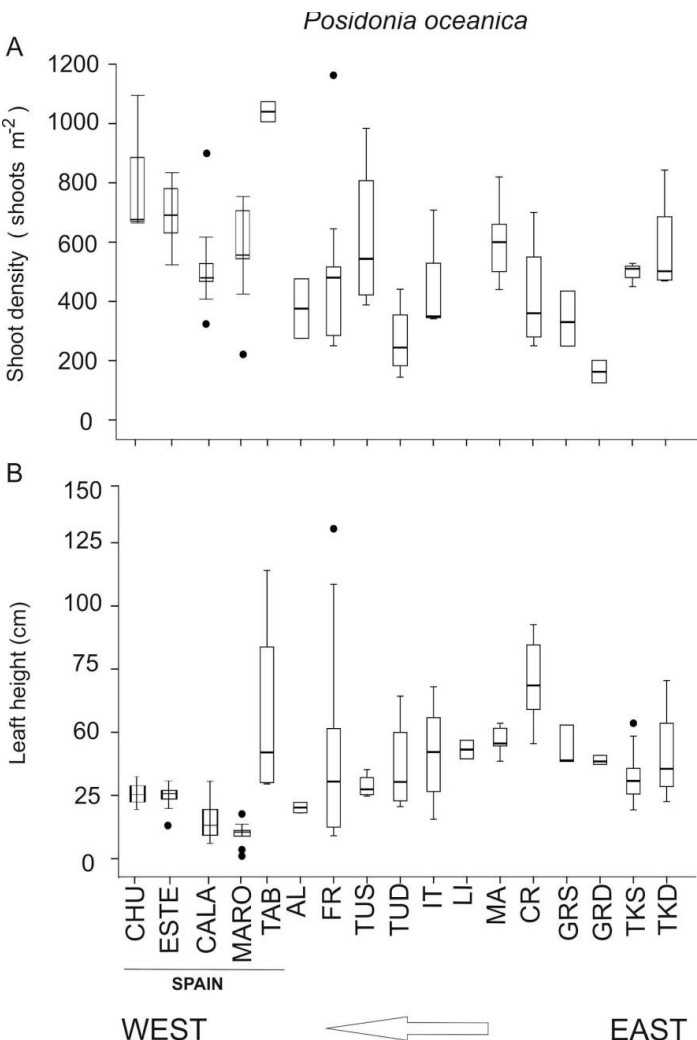

**Figure 6.** Shoot density (**A**) and leaf height (**B**) of *Posidonia oceanica* at the studied sites of the northwestern Alboran Sea (in the case of Calahonda SAC, data were clustered from different seasons and years considering Calaburras and Calahonda), as well as at other locations of the Mediterranean Sea. The bold black line in the box represents the median value. The upper and lower ends of the box represent the third ($Q_3$) and first ($Q_1$) quartiles, respectively. The ends of the lines extending vertically from the bars (whiskers) are the minimum and maximum values. Outliers are plotted as individual points. CHU, Punta Chullera; ESTE, Estepona SAC; CALA, Calahonda SAC; MARO, Maro MPA; TAB, Tabarca Island (E Spain) [32]; AL, Algeria [77]; FR, France [30,34]; TUS and TUD, Tunisian shallow (<5 m depth) and deep (>5 m) meadows, respectively [73]; IT, Italy [31,33,36]; LI, Libya [78]; MA, Malta [79]; CR, Croatia [50]; GRS and GRD, Greek shallow (<5 m) and deep (>5 m) meadows, respectively [49]; TKS and TKD, Turkish shallow (<5 m) and deep (>5 m) meadows, respectively [12].

Disentangling natural, anthropogenic impacts on different organisms and habitats is still very difficult to achieve. In spite of being exposed to high anthropogenic pressure (intense coastal development and high human population), most of the studied *P. oceanica* meadows were in a good conservation status as reflected in the BiPo index, with EQR values similar to those of other meadows located in areas that are more pristine and where the human population density is low [55]. This could be related to the peculiar characteristics of the *P. oceanica* meadows at our study sites compared to other sites in the Mediterranean Sea (see Figure 6). Nevertheless, the Maro MPA meadow showed evidence of decline, which could be partly related to large flooding linked to a torrential rainfall (200 mm per m$^2$) recorded in the area in September 2007. This had serious consequences for the *P. oceanica* meadow located here as several streams flow into the small cove of "Molino

de Papel". Flooding of these watercourses led to the transport of sand and mud, plant debris, and waste of different kinds into the sea, which were deposited in a large part of the meadow. This caused the burial and subsequent death of approximately half the existing meadow [89] (see Figure 2D). The Calahonda SAC meadows also show a high presence of old and dead rhizomes, with no leaves and the matte covered by macroalgae, which may indicate that *P. oceanica* has suffered a regression and it was more abundant in the past (see Figure 2C).

## 5. Conclusions

The *P. oceanica* meadows of the northwestern Alboran Sea represent the westernmost located ones ever studied. These meadows have an important ecological role in the area, supporting high-diversity communities [4,5,14,15,43,44,59] and showing some unique characteristics (e.g., higher shoot density and lower leaf height) that differentiate them from other Mediterranean meadows present at similar depths. Nevertheless, the seasonal dynamics of phenological parameters were similar to those recorded in other areas that fall within the biogeographical distribution of *P. oceanica*. Although the conservation status of the studied meadows is good and flowering events have been reported, the present study highlights recent impacts on some of them linked to different anthropogenic pressures occurring at the studied coastal area of southern Spain (e.g., coastal infrastructures, elevated urbanism, insufficient wastewater treatment plants and illegal fishing). For these reasons, it is recommended to carry out detailed long-term monitoring of these meadows in order to improve the management of the westernmost populations of this EU-priority habitat. Further monitoring of these seagrass meadows as part of the Marine Strategy Framework Directive may provide crucial information on the improvement or decline in the state of these singular meadows located in a very different environmental situation compared to other Mediterranean ones. Furthermore, studies on the potential restoration of *P. oceanica*, and other seagrasses, will contribute to balancing the alarming and continuous regression of meadows detected in the northern Alboran Sea during the last decades [90].

**Supplementary Materials:** The following are available online at https://www.mdpi.com/article/10.3390/oceans4010003/s1, Table S1: Two-way PERMANOVA pair-wise analysis of *P. oceanica* parameters among months and years.

**Author Contributions:** Á.M.-R., P.M., J.L.R. and J.U. contributed to the conceptualization and design of the study. Á.M.-R., P.M., J.L.R., E.B.-E. and J.U. contributed to data acquisition. Á.M.-R., P.M., A.M.-A., E.B.-E., J.E.G.R., J.L.R. and J.U. contributed to writing, editing, and reviewing the manuscript. J.L.R. and J.U. supervised and coordinated the study. All authors have read and agreed to the published version of the manuscript.

**Funding:** This research received no funding.

**Institutional Review Board Statement:** Not applicable.

**Acknowledgments:** We would like to express our sincere gratitude to the "Consejería de Medio Ambiente de la Junta de Andalucía" and RNM-0141 Research Group from the Universidad de Málaga for their institutional support, the staff of the "Centro de Buceo Benalmádena", for their help and continuous interest, and to Terence W. Edwards from Sunny View School for the English revision of earlier drafts of this manuscript.

**Conflicts of Interest:** The authors declare no conflict of interest.

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
