# Peer review of "Posidonia oceanica (L.) Delile at Its Westernmost Biogeographical Limit (Northwestern Alboran Sea): Meadow Features and Plant Phenology"

_2673-1924, doi:10.3390/oceans4010003_

Round 1

Reviewer 1 Report

The paper by Ángel Mateo-Ramírez et al. 

Posidonia oceanica (L.) Delile at its westernmost biogeographical limit (north-western Alboran Sea): Meadows characterisation, phenology and flowering 

is a descriptive study on the meadow features and most of the plant phenology of Posidonia oceanica settlements at its westernmost distribution limit in the Mediterranean Sea (Albora Sea).  The study report many data on the studied meadows, is conducted on a medium-term temporal basis (several yeas at least in two stations) and include also flowering events. Therefore, due to the important geographic location of these Posidonia meadows, these data deserve publication...

I found some confusion in a few statements  on some parameters discussed (shoot density and leaf height...) which can be better explained and clarified, as well as for flowering quantification.... I am surprised also for high density values of flower occurrence, and this could be better discussed compared to other Posidonia meadows, and possibly using also an additional metrics (e.g., the % of flowering should be used, according with Pergent Martini & Pergent, if you have shoot density data you can easily calculate)....

For the rest and  most of the text the data presented are clear and well supported by statistics and well discussed....

I wonder why the Authors did not performed the ledidochronological analysis to generate further information on the past natural history and producion features  on these interesting meadows ....this is a suggestion for the future....

the paper require only modest clarification, and I have included all my comments directly in the pdf of the Ms ad notes and suggestion/corrections...nd in my opion should be published in the Journal...

with my kindest regards,

 reviewer 1 

Author Response

We added the document with our responses.

Reviewer 2 Report

The manuscript overall presents valuable information and should be published; acceptance in the present journal is recommended.

My main reservations are as follows:

(i) The English could be better; I have tried to indicate where this can be improved but review of the language by someone fluent in writing in English is desirable.

(ii) The authors would do well to clarify use of 'site' and 'station', since I found this unclear in places. To me the sites are Maro, Calahonda, Chullera etc and stations are points of sampling within each of these.

(iii) The authors would do well to check interpretation of results of shoot density variation (Line 476 onwards). Was any vaiation of this attribute seasonal (intra-annual) or amongst years (inter-annual) or both? If the former, it would seem unusual.

I have included suggestions and comments on the ms itself.

Author Response

(The authors gave the same response as above.)

Reviewer 3 Report

This is a descriptive study of seagrass populations near their western limit I the Mediterranean Sea. Although the authors do a good job of contextualizing their results within the wider Mediteranean the Discussion is very long and a little repetitive in some places. If it could be reduced it would be better to read.

So a general point is to try to remove repetition of Results in the discussion and keep the discussion very honed. I think Table 7 and text should be moved to Results because it is results and briefly discussed in the discussion.

Other points:

Please remove or improve the last sentence of the abstract which reads like a conference abstract written before all the results were available.

 Line 75 and 409 suPrficial

Figure 1 needs to be re-lettered A-E so that the map is A and referede to

Line 131 replace higher with greater

Lne 142 As SCUBA is an acronym (Self Contained Underwater Breathing Apparatus) “this should just be “was done using SCUBA-diving during....”

Line 153 and throughout replace quadrant with quadrat (a quadrant is ¼ of an area).

Line 227 determinate – determined.

Table 1 and Figure 5 heigth typo

Author Response

(The authors gave the same response as above.)

Reviewer 4 Report

I have three comments and some minor spelling issues.

1) In the introduction (e.g., lines L 88:”…studies on phenological aspects of P. oceanica meadows are very scarce at the limits of its distribution range...” and again in the discussion (L 539-542) you bring up the issue of monitoring an edge populations. There is long list of literature that comes from terrestrial plants and highlights the importance of populations at the edge of a species' spatial distribution. These studies highlight the importance of such “peripheral” or “edge” populations for a species' long-term survival and evolution. I think you touch on this but could easiliy go deeper. 

even more: Considering the potential effect of climate change on species, whereby populations move north or south just to stay in the “same” climate regime, edge populations are of even more importance.

Checkout these two examples

-        Lesica P, Allendorf FW. When Are Peripheral Populations Valuable for Conservation? Conserv Biol. 1995; 9(4):753±60.

-        Gibson SY, Van Der Marel RC, Starzomski BM. Climate change and conservation of leading-edge peripheral populations. Conserv Biol. 2009;

 Again, this text comes from terrestrial plants – but they are the same for your work..

declines in populations located at a species' edge may restrict possible range expansion.. such a species in the near future. These concerns call for the protection of peripheral populations. 

2) Methods:

L 198 – 200 : “Solar irradiance measurements were obtained using HOBO Data Loggers that were 198 deployed seasonally from winter 2009 to winter 2010 (an annual cycle). Each HOBO was placed daily close to the sampled seagrass meadows (at 4-5 m depth) and set to record 200 data with 10 min intervals between 10:00 and 14:00.”

this is MIDDAY solar irradiance,  Which HOBo you use – model ?

 L150: At each site, patches of different sizes were sampled, including small patches (patch size <1 m2; n= 3), medium patches (1-5 m2; n= 3) and large patches (>5 m2; n= 3).

Three quadrants of 50×50 cm were deployed for shoot density measurements in each of these patches (n total= 27 measurements per site).

While I understand you have 3 patch sizes, and in each size patch you have three replicated same size patches, and in each one of those you lay 3 50x50 quadrants…., so while you laid a total of 27 quadrates , your n is not 27….it is still n=3….

Same goes for shoots – not 270 but n=10….

L180: n = 50 shoots per season and sampling station – yet again, you have  5 quadrants (50×50 cm) at each season and sampling station” so it does not matter how many shoots you count per site, your n is still n=5….

3) Results: 

Since in the methods (lines 143-144) you list : “…substrate types and bathymetric ranges where P. oceanica occurs, its coverage, shoot density and some phenological parameter”

The reporting of the results should follow the same order – first report of data should be on the substrate type.

Author Response

(The authors gave the same response as above.)

Round 2

Reviewer 1 Report

After revision by the Authors,  the Ms is now ready to be published in the present form.

Author Response

Thank you for your comments.

Reviewer 2 Report

The revised ms is a significant improvement on the first version; congratulations to the authors! However, I believe that some aspects of the English text can still be improved, noting that scientific writing is different from classical English writing. Accordingly I have tried to indicate (see below) where improvement can be made.

There is one item in the methodology section (measurement of shoot density in small seagrass patches <1m2, which the authors would do well to address.

ABSTRACT

Remove ‘acute’ from penultimate sentence.

Replace ‘few’ by ‘some’.

INTRODUCTION

First para

In the sentence starting “These meadows support……” I suggest revision of “…, buffering of the coastal waters quality, coastline protection, nutrient ….” to “…., buffering of strong water movement leading to protection of the coastline against erosion, nutrient….”.

Second para

Remove second “the” from first sentence.

Remove “the” from second sentence.

Remove first and second “the” from third sentence.

Third para

Replace “This seagrass” by “Posidonia oceanica”.

Fifth para

Sentence “In this line, peripheral or edge populations are important for a species long-term survival and evolution [36,37], as these populations usually present a high genetic differentiation compared to central populations due to reduced population size and genetic drift in fragmented habitats, which result in a lower genetic diversity [38].” Does not read well. I suggest revision to “WITHIN THIS CONTEXT, peripheral or edge populations are important for a species’ (NOTE INSERTION OF APOSTROPHE HERE) long-term survival and evolution [36,37], as these populations usually present a high genetic differentiation compared to central populations due to reduced population size and genetic drift in fragmented habitats, which result in a lower genetic diversity [38]."

Place “shocks” in singular and remove “it” from “…their response depends on local population traits, as it has been observed in some marine macrophytes (e.g. Laminaria digitata, Fucus vesiculosus, Zostera marina, P. oceanica) [39-41].

MATERIAL AND METHODS

Revise “This site covers 15.29 km2 of MPA with a depth range from the coast down to ca. 70 m.” to “This site covers 15.29 km2 of THE MPA with a depth range from the SHORE down to ca. 70 m.

In the following sentence, replacing “coastline” with “shore” would seem better.

In their covering letter the authors respond to my query “If some of the patches were <1m2 and others just over 1m2, how could 3 replicate shoot density counts using a 50 cm x 50 cm quadrat be made?!” by stating “In that specific case, we made one of the replicates (= quadrat) in the most adjacent patch.” However, the authors do not indicate this in their ms, when they should do so as this is an important methodological discrepancy.

Para under the sub heading “Statistical analyses”. I think the authors mean “raw data” not “row data”.

Caption to Figure 2. I suggest inclusion of “Posidonia oceanica” at the start; i.e. “Figure 2. (A) Posidonia oceanica patch with several….”

DISCUSSION

First para.

Where used, I suggest replacing “In this line….” By “Within this context…” OR “In this respect….” or remove this phrase completely. This also applies to the second para of the discussion.

Replace “examined” by “studied”.

Author Response

Dear reviewer 2, 

You can find the attached response letter here. 

Reviewer 3 Report

No further comments

Author Response

Thank you.

Reviewer 4 Report

The paper is well written and deserves to published. But I have some comments. Their were no line numbers in the version I downloaded from the website, so I wrote which sections I had comments on:

1. Introduction:

“In this line, peripheral or edge populations are important for a species long-term survival and evolution [36,37], as these populations usually present a high genetic differentiation compared to central populations due to reduced population size and genetic drift in fragmented habitats, which result in a lower genetic diversity [38]. Ecotypes derived from this genetic differentiation respond differently to environmental changes, being able to show even better tolerance to thermal shocks than central populations

You do not provide any citations for this very strong sentence.

2. "providing an elevated number of ecosystem services such as the maintenance of marine biodiversity (including species of commercial interest), buffering of the coastal waters quality, coastline protection, nutrient cycling and carbon sequestration, among others  (2,4,5) " - two of these are reports and not published papers. The 3rd paper is a much better choice. But I would have chosen here much more classic "big shot" references 

3. Figure 1A. Map of the study area in the north-western Alboran Sea and images of the general configuration of Posidonia oceanica patches in the studied meadows. Orange areas indicate the location of the different sites with P. oceanica meadows: (A) Maro MPA, (B) Calahonda SAC, (C) Estepona SAC and (D) Chullera. Green lines indicate areas influenced by nutrient-rich upwellings. Yellow arrows indicate the direction of the Surface Atlantic Waters and pink arrows indicate the direction of the Mediterranean Intermediate and Deep Waters

- ON WHAT SOURCE/PAPER are these YELLOW/GREEN LINEs BASED ON ? I am sure it is information that is well known, but please cite a source/paper

5. Statistics

I confess that I am not the strongest statistical person in the world, but I have big doubts about the way the stats model used. The authors used two different permanovas:

1. A two-way PERMANOVA design 1 with two fixed factors: site (4 levels: Maro MPA, Calahonda SAC, Estepona SAC, Chullera) and patch size (3 levels: small, medium, large).

2. A two-way PERMANOVA  design 2 to test for statistical differences of environmental variables between seasons and years.

But as far as I understand, we have monitoring at have different sites  (4 sites), years (6 years), seasons (4 per year), and different patch sizes (small, big, medium). Not to mention that you have at each site different depths, and  different substrates.

I can’t see any model/fig/table taking into account a possible interaction of more than 2 parameters of this list, not to mention an interaction between ALL of these (site, year, season, depth, patch size).  Also meadows had a gradient in the meadow structure from continuous meadows to small patches towards the SoG - where is this in the model?

I am not sure the supp table answers these potential interactions...

Author Response

Dear reviewer 4

You can find the attached response letter here. 
